# Allred Scoring of ER-IHC Stained Whole-Slide Images for Hormone Receptor Status in Breast Carcinoma

**DOI:** 10.3390/diagnostics12123093

**Published:** 2022-12-08

**Authors:** Mohammad Faizal Ahmad Fauzi, Wan Siti Halimatul Munirah Wan Ahmad, Mohammad Fareed Jamaluddin, Jenny Tung Hiong Lee, See Yee Khor, Lai Meng Looi, Fazly Salleh Abas, Nouar Aldahoul

**Affiliations:** 1Faculty of Engineering, Multimedia University, Cyberjaya 63100, Selangor, Malaysia; 2Department of Pathology, Sarawak General Hospital, Kuching 93586, Sarawak, Malaysia; 3Department of Pathology, Queen Elizabeth Hospital, Kota Kinabalu 88200, Sabah, Malaysia; 4Department of Pathology, University Malaya Medical Center, Kuala Lumpur 59100, Malaysia; 5Faculty of Engineering and Technology, Multimedia University, Ayer Keroh 75450, Melaka, Malaysia

**Keywords:** Allred scoring, estrogen receptor, hormone receptor, tumor biomarker, breast carcinoma, digital pathology

## Abstract

Hormone receptor status is determined primarily to identify breast cancer patients who may benefit from hormonal therapy. The current clinical practice for the testing using either Allred score or H-score is still based on laborious manual counting and estimation of the amount and intensity of positively stained cancer cells in immunohistochemistry (IHC)-stained slides. This work integrates cell detection and classification workflow for breast carcinoma estrogen receptor (ER)-IHC-stained images and presents an automated evaluation system. The system first detects all cells within the specific regions and classifies them into negatively, weakly, moderately, and strongly stained, followed by Allred scoring for ER status evaluation. The generated Allred score relies heavily on accurate cell detection and classification and is compared against pathologists’ manual estimation. Experiments on 40 whole-slide images show 82.5% agreement on hormonal treatment recommendation, which we believe could be further improved with an advanced learning model and enhancement to address the cases with 0% ER status. This promising system can automate the exhaustive exercise to provide fast and reliable assistance to pathologists and medical personnel. The system has the potential to improve the overall standards of prognostic reporting for cancer patients, benefiting pathologists, patients, and also the public at large.

## 1. Introduction

Breast cancer is the most common cancer occurring in women and is the second leading cause of cancer-related deaths in women. The majority of breast tumors and breast cancers are first detected using either mammography, magnetic resonance imaging (MRI), or ultrasound scans. However, for better diagnosis and prognosis of breast cancer, tissue samples must be obtained, either through biopsy or surgery, for analysis by pathologists. In the case of breast cancer treatment, a crucial step is to test the tumor tissue to determine if it has estrogen receptors (ER), progesterone receptors (PR), and/or human epidermal growth factor receptor 2 (HER2). These markers provide key information about how the cancer may behave. Along with tumor grade and cancer stage, tumor marker status helps determine the best treatment options for breast cancer patients. ER, together with PR, has been recognized as a “predictive” marker for which women with breast cancer would respond to hormonal treatment.

Predictive immunohistochemistry (IHC) is commonly used in breast histopathology practice to determine the expression of hormone receptor proteins. The use of IHC to assess the ER and PR status of breast cancers in formalin-fixed, paraffin-embedded (FFPE) tissue sections of cancer samples is now a routine part of pathology practice worldwide and is recommended to be performed in all primary invasive breast carcinomas and on recurrent or metastatic tumors. The hormone receptor status of a breast cancer helps the doctor to decide whether the patient should be offered hormonal therapy or other treatments. Hormonal therapy includes medications that either lower the amount of estrogen in the body or block estrogen from supporting the growth and function of breast cells. If the breast cancer cells have hormone receptors, then these medications could help to slow or even stop their growth. Patients with ER-positive cancers are highly receptive to endocrine therapy and have a higher chance of survival [1]. If the cancer is hormone-receptor-negative (no hormone receptors are present), then hormonal therapy is unlikely to work. In this case, other kinds of treatment should be sought.

Valid determination of ER and PR status is thus a prerequisite for establishing adequate treatment strategies for breast cancer patients, regardless of disease stage. The determination of these protein expressions, however, is currently carried out manually, which is not only tedious and time-consuming for the pathologists, but is also prone to errors and inaccuracies. In this paper, we proposed a system for automated estrogen receptor status evaluation in breast carcinoma patients which consists of four stages: cell detection, positive/negative classification, weak/moderate/strong classification, and Allred scoring. As both ER and PR share similar staining characteristics, the system can be further extended to determine PR (and possibly HER2) expression in the future with minor modifications to make for a complete hormone receptor system.

Figure 1 shows the block diagram of our proposed system. In our previous work, indicated by blue boxes and arrows, we have reported our initial work on cell detection [2], positively and negatively stained (PN) cell classification [3] and weakly, moderately, or strongly stained (WMS) cell classification [4] on their own. In this work, we integrated the three stages into a single system, experimented with each stage with the same set of images, and evaluated their performances objectively and transitively in order to observe the overall performance of the integrated system. Positively and negatively stained cells are quite distinguishable based on their color properties, i.e., negative cells are stained with a blue/purple hue, while positive cells are stained with a brown hue. Color analysis based on the weighted hue and value of the cells (from the HSV color model), which we previously used in p53 expression analysis of brain glioblastoma, can be used.

Cell detection and WMS classification on the other hand, are more challenging. For cell detection, the challenges lie in detecting the boundaries, in which some cells which are too close to each other appear as if they belong to single large cells, while some others have rather weak and unclear boundaries. For WMS classification, the differences between the weakly, moderately, and strongly stained cells are not very obvious. The moderately stained cells especially are very tricky as they can easily be classified into either the weak or strong classes. The convolutional neural network (CNN) excels in these types of challenges by learning the features end to end while avoiding the manual feature selection in traditional image classification. Because of this we decided to use a deep convolutional neural network for both the cell detection and WMS classification stages. Cell detection was carried out using CNN with regression layer in order to obtain accurate cell boundaries (hence correct cell detection), while WMS classification utilizes the CNN with the regular classification layer to classify the cells into the three staining strengths.

The detected and classified cells are then fed to the final stage of the system, indicated by the red box and arrows, for estrogen receptor status evaluation. By computing the distribution of the strong, moderate, weak, as well as negatively stained cells within the slides or regions of interest, the estrogen receptor expression for the slide is determined by computing the Allred score [5], and compared to the pathologists’ manual scoring. Alternatively, the H-score [6] can also be used for assessing estrogen receptor status, but in this project we focus on Allred scoring as we have the ground truth available for evaluation. To the best of our knowledge, this is the first work of its kind in developing such a system with the scoring of whole-slide images, which would be a valuable tool for histopathologists in improving the reliability of tumor marker reporting as well as reducing manual intervention workload. There are some similar works found in the literature [7,8,9] working on Allred scoring of ER-IHC patches. Since we aim to replicate the pathologists’ process flow in deciding the score, which is on whole-slide images, the existing work from the literature is not a fair comparison either in terms of accuracy or computational time.

This paper is organized as follows. Section 2 describes all the stages involved in the proposed automated Allred scoring system in detail. Section 3 explains the experimental set-up, while Section 4 discusses the experimental observations. Finally, the conclusion and future works are presented in Section 5.

## 2. Methods

In this section, the three prior stages to the Allred scoring, which are cell detection, PN classification, and WMS classification will be described before discussing the Allred scoring methodology for hormone receptor evaluation.

### 2.1. Cell Detection

Our cell detection stage is based on the work by [10], where a score map is predicted based on the Euclidean distance transform for cell centers in a given input image. The prediction of the score map is performed using a regression method trained from cell images and ground truth cell locations, similar to the work by [11]. The idea is to train the network to fine-tune the boundary of individual cells regardless of their class, shape, color, and overlapping parts, if any. In constructing the model, we experimented with a network up to 20 layers by stacking the convolution, rectified linear units (ReLU), and max pooling (maxpool) layers, followed by the regression layer at the end. Due to the limitation of our workstation, we kept the design minimal and the footprint smaller. Figure 2 shows the proposed CNN model for our cell detection stage, together with an example of the score map.

During training, we asked our collaborating pathologists to mark each cell inside small regions of around 500 × 500 pixels as the ground truth. Each cell mark is expanded as a circle with a radius of 32 pixels, and 64 × 64 patches were extracted for use to train the network. During detection, a score map is generated for the whole image by the CNN-based regression model, and initial segmentation is obtained by thresholding the score map (a score of 0.2 and higher constitutes cells). To address the problem of closely connected cells, watershed-based boundary processing is carried out. Another thresholding is then applied to remove the non-cells region (an area greater than 240) from the final detection. As we are more interested in cell detection rather than cell segmentation, the choice of thresholds does not affect the final outcome much. The centroids of the detected cells are passed to the next stage for classification.

### 2.2. PN Classification

The classification of cells into positively and negatively stained cells is based on our earlier work in classifying positive and negative cells in p53 expressions [3,12] of brain glioblastoma. Similar to p53 images, the positively and negatively stained cells in ER-IHC stained images are quite distinguishable based on their color. Negative cells are characterized by their blueish hue and low intensity. Positive cells, on the other hand, are characterized by their brownish hue and rather high intensity. The classification of the cells into positive or negative stains can thus be based on the intensity and color of the cells.

For each of the detected cells found in the previous stage, 32 × 32 pixel blocks are extracted around their centroid, and converted into the HSV (hue/saturation/value) color model. While most cells fit nicely into this 32 × 32 pixel block, there are some cases where the cells are too big or too small. For the former, we should still be able to obtain the color and intensity information from the part of the cells that fit into the block. For the latter, however, it is possible that some other cells may also be captured by the block, thus compromising the color and intensity information of that particular cell. To address this, we used weighted hue and weighted value instead to compute the color and intensity, respectively. The weights used are inversely proportional to the pixels’ distance to the centroid, with those closer to the center of the block receiving higher weight, and those further from the center receiving less weight.

The weighted hue and value are calculated for each block and these values are used for classifying the cells. Negative ER-stained cells tend to be blue (higher hue) and less intense (higher value), while positive ER-stained cells tend to be brown (lower hue) with varying intensities. Based on these properties, we propose a two-step classification rule: (1) if the weighted value (wV) for a block is less than a particular threshold (darker), the block will be classified as containing a positive cell, regardless of its weighted hue (wH); (2) otherwise, the classification depends on weighted hue, with wH less than a particular threshold meaning that the block contains positive cells, and wH more than the threshold meaning it contains negative cells. From experiment, wH > 40 and wV < 50 were found to be suitable thresholds for our ER-IHC images. Figure 3 summarizes the proposed positive/negative cell classification process in a flowchart.

### 2.3. WMS Classification

In order to get the best result in learning the features of ER-positive cells, a different CNN model is proposed, which is based on earlier CNNs such as LeNet [13], ImageNet [14], GoogLeNet [15], and a few others [10,11,16,17], and their recommendations. These include the effect of the convolutional network depth on its accuracy in large-scale image recognition settings; usage of back-to-back convolution layers with padding to maintain more pixel information in shrinking spatial information but increase model layer depth; usage of ReLu as the activation layer to reduce training time, and the addition of dropout layer to reduce overfitting in imbalance dataset. Nevertheless, similar to the model used in cell detection, the key is to be able to run an optimized CNN model with a small footprint using our limited hardware capability. Our network for this 3-class classification problem is made up of nine convolutional layers including the fully connected layer. Figure 4 shows the proposed CNN model for the WMS classification stage. The input to the network is 32 × 32 patches of positive cells from the previous classification stage, while the output is one of the strong, moderate, or weak classes.

The same regions used for the cell detection experiment were used in generating training and testing samples and their ground truth. Data augmentation such as horizontal flipping, random cropping, and normalization is done to increase our training samples. The network is trained for 600 epochs using Matconvnet [18]. The weight of each layer is initialized by multiplying a small random number with the zero mean, sampled from Gaussian’s standard deviation. Stochastic gradient descent is used to reduce the objective loss with a learning rate set in two vectors of logarithmic space: the first 10 epochs are set to 10-2 and the remaining epochs are set to 10-4 increments to improve the network convergence.

### 2.4. Allred Scoring

In this work, we used the Allred score to express ER, where the evaluation is based on the proportion of ER-positive cancer cells and the intensity of the reaction product in most of the positive cells [19]. Table 1 shows how the respective proportion and intensity scores are derived. Based on the percentage of ER-positive cells, the cancer is assigned one of six possible proportion scores (0 to 5). Based on the intensity of most of the ER-positive cancer cells, the cancer is also assigned one of four possible intensity scores (score of 0 to 3 for negative, weak, moderate, and strong, respectively). The 2 scores are then added together for a final score with 8 possible values (the Allred score). Allred scores of 0 and 2 are considered negative for ER (i.e., not actionable), while scores of 3 to 8 are considered positive (i.e., recommended for hormonal therapy). Note that a score of 1 is not a possible outcome.

Given an ER-stained whole-slide image, all the cells in the image are first detected and classified into one of 4 classes: negatively, weakly, moderately, and strongly stained cells (N, W, M, and S, respectively), as described in the previous three stages. The percentage of positively stained cells over all cells, as well as the intensity score derived from the majority class, is then used in computing the Allred score for hormone receptor status for the particular slide.

## 3. Experimental Setup

Our ER-stained whole-slide images (WSI) of breast cancer are scanned using a 3DHistech scanner at 20× magnification with a resolution of 0.243 micrometer/pixel, resulting in images with a resolution of more than 80,000 by 200,000 pixels. Altogether, 40 whole-slide ER-stained images are available for use in all our experiments (refer Table 2). All 40 whole-slide images were uploaded to our in-house Linux lab server which has been installed with web interface [20] as shown in Figure 5, enabling an easy online WSI viewing and annotation for the pathologists marking the ground truth. The ground truth for the images (cell location, class, Allred score, etc.) is provided by pathologists from the Department of Pathology, University of Malaya Medical Center.

For cell detection and classification, the evaluation is based on the ability of the system to detect and classify cells. For this, comprehensive annotation by the collaborating pathologists was required. Since it is impossible to annotate all cells in the whole-slide images, the following approach is observed. For each of the 40 whole-slide images, a small region (around 500 × 500 pixels) is selected for annotation by the pathologists. However, due to time constraints, only 37 regions were eventually annotated. Thus 37 regions will be used in the detection and classification experiments: 30 regions used for training and 7 regions used for testing. For a particular region to be annotated, the pathologists were asked to identify all cells, and label them into either negatively, weakly, moderately, or strongly stained cells. Cells were annotated by marking the center of the cell, followed by annotating the cells into one of the four classes.

Figure 6 shows a couple of examples for the 37 regions used for this experiment, together with their annotation. A total of 3445 cells were marked by the pathologists from 37 annotated regions. Two-fold cross-validation was used, and we refer to these experiments as Experiments A and B, respectively, as detailed in Table 2. All the marked cells were later extracted as 32 × 32 augmented image patches. Data augmentation such as horizontal flipping, random cropping, and normalization is carried out to increase our training samples. To evaluate the performance of the system in detecting and classifying the cells, the number of true positives (TPs), false positives (FPs), true negatives (TNs), and false positives (FPs) were recorded. The performance of the proposed classification system is presented in terms of the precision and recall for each of the detection, classification, and quantification.

For the Allred scoring experiment, the evaluation is done at the whole-slide level, i.e., to compare the Allred score automatically computed by the system to the ones estimated by the pathologists. No detailed cell-level annotation is required, instead only the Allred score from the pathologists’ calculation/estimation for the 40 images is needed as the ground truth. Note that even though for 30 of the image there is some bias (since there is a region within the slides that were used in training), these regions are very small (around 500 × 500) compared to the whole slides (less than 0.01%), so the bias is very minimal, if any. To evaluate the performance of the Allred scoring, cell detection and classification are applied on each of the 40 whole-slide images (divided into non-overlapping high power field regions within the tissue area), before Allred scoring is carried out for each image and compared to the ones manually computed/estimated by the pathologists. Figure 7 shows one example of the 40 whole-slide images used in this experiment.

## 4. Results and Discussion

The results are presented in two subsections. In the first subsection, the performances of cell detection, PN classification, and WMS classification are evaluated at the cell level objectively and transitively. In the second subsection the performance of Allred scoring is evaluated at the slide level.

### 4.1. Cell Detection and Classification

As mentioned in the previous section, 30 regions were used for training and 7 images were used for testing. Two-fold cross-validation was carried out, named Exp A and Exp B, respectively. Table 3 shows the true positives, false positives, false negatives, recall, and precision for cell detection results for each of the 14 test images from the two-fold experiments. Overall there are 1425 cells from the 14 test images, 1328 of which were correctly detected. A total of 97 cells were missed, while 62 non-cells were detected as cells, giving a precision of 93% and recall of 96%, which shows that the proposed system is able to carry out its task reliably. Upon closer inspection, of the 62 false positive cells, 46 are non-tumor cells of the supportive stroma such as stromal cells and epithelial cells, and only 16 are actually non-cells. An improved system that can differentiate between tumor cells from stromal and epithelial cells would further improve the detection performance.

Table 4 shows the summary of the PN classification and Table 5 shows the classification of the detected cells into positive and negative cells for each of the 14 test images. The 62 false positive cases (non-tumor cells and non-cells wrongly detected in the previous stage) are also included in the classification experiment (last row of Table 3). As can be seen, only 13 cells were wrongly classified among 1328 correctly detected cells from the previous stage. The overall accuracy for all the detected cells is 95%, but if the false positive cases are removed (the error actually stemmed from the previous stage), the accuracy improved to 99%. This is also true for the true positive rate (correct classification of positive cells) and true negative rate (correct classification of negative cells). Overall, the proposed color-based classification is very reliable in classifying cells into positively and negatively stained cells.

Table 6 shows the classification of the detected positive cells into the strong, moderate, and weak classes for the 14 test images. The negative and non-tumor cells wrongly detected and classified as positive cells in the previous stages are also included in this experiment. As can be seen, most cells are classified correctly. Of the incorrect classification, 10 are from strong cells (classified as moderate), 13 are from moderate cells (6 classified as strong, and 7 as weak), and only 1 from weak cells (classified as moderate). Most of the negative and non-tumor cells that were classified as positive are classified as weak. Table 7 summarizes the overall accuracy as well as the positive prediction value (PPV) and the true positive rate (TPR) for each class. The overall accuracy of 88% is observed when including the false positive cells from prior stages, which improved to 90% if these cells were excluded.

The performance of the WMS classification agrees with our previous finding on 1200 extracted individual cells (400 weakly, 400 moderately, and 400 strongly stained cells). As reported in our previous article [4], 1066 out of 1200 cells have been classified correctly, which constitutes 88.8% accuracy on average and an overall AUC (area under curve) of 97.5%. The individual PPVs for the strong, moderate, and weak categories were reported as 90.5%, 88%, and 88%, respectively, which is in line with what is observed in this experiment. The results from this experiment on 14 test regions, as well as the previous experiments on 1200 individual cells (32 × 32 blocks) proved that the proposed WMS classification algorithm is reliable in classifying the positive cells into the three staining strengths. Figure 8 shows several visual examples of the final detection and classification results, while Figure 9 shows the proposed algorithms integrated into our ER-IHC breast carcinoma assessment system, for use in prognostic applications.

### 4.2. Allred Scoring

Table 8 shows the computer prediction of the Allred scoring for all 40 whole-slide images against the pathologist’s estimation. The images were sorted according to the pathologist’s estimated Allred score, from lowest to highest. As mentioned in the previous section, there are eight possible scores for each slide, with scores of 0 and 2 considered negative, while scores of 3 to 8 are considered positive (actionable for hormonal treatment). For more than half of the images (23), the computer predicted exactly the same score as the pathologist’s estimation. Seven other images recorded a difference of only a single score, while another seven images recorded a difference of two scores. Only three images recorded a difference of more than two between the pathologist’s estimation and the computer prediction. Figure 10 illustrates the difference better, where it can be seen that except for a few images, the computer prediction more or less follows the general pattern of the pathologist’s estimation.

It is interesting to note that for the four images considered to be 0% ER status by the pathologists (i.e., 100% of the cells were stained negatively), none of them were correctly predicted by the computer. The reason is, while the computer detection more or less agrees that the cells are almost entirely negative, it still detected a very small percentage of brownish objects, which could be either actual positive cells or other artifacts with similar characteristics to the positive cells such as noise. These images will thus automatically have a score of at least 1 for the proportional score (a score of 1 for those between 0% and 1%, and a score of 2 for those between 1% and 2%), and a score of at least 1 for the intensity score (since the intensity score is still given regardless of how small the positive proportion is). The final Allred scores for these four images are 2, 2, 3, and 4, respectively. In future work, we will work with our collaborating pathologists to suggest possible rules for a more accurate prediction for this kind of images.

It is possible that the pathologist misses some positive cells due to the sheer size of the images, so the rule may be very useful. The only other image with a score difference of more than 2 is image 05441, where the pathologists estimated an ER status of less than 1%, while the computer predicted it to be close to 50%. Upon detailed inspection, it was found that this particular image was a rather brownish in general compared to other images with similar status, which could be caused by an error during the staining process. Overall, the proposed approach managed to predict the scores similar to the pathologists, except for a few cases explained earlier.

In terms of hormonal treatment (not actionable vs. recommended for hormonal therapy), 33 out of 40 computer-predicted treatments agree with the pathologist’s recommendation, giving an accuracy of 82.5%. Note that the pathologist’s recommendation is based on manual computation and/or estimation from sampled regions, while the computer prediction is based on the counting throughout the whole slides. It is possible that there may be some bias in the pathologist’s sampling, although this will be very hard to prove due to the large size of the images. The computer prediction, on the other hand, is free from this sampling bias, as all regions are considered. Overall this is a very promising result. Out of the seven disagreements, four are caused by the single difference between scores 2 and 3, which can be addressed with a more advanced learning model. The other three were from the 0% ER status and staining error case, as discussed previously. It is also interesting to note that five out of the seven disagreements are for the negative treatment cases, meaning the system has more difficulties predicting negative treatment cases. Further improvement to the deep learning model can help to address this problem.

The results demonstrate that the proposed system can be very useful in assisting pathologists in their predictive tumor marker reporting of breast carcinoma. It helps to reduce the bias in sampling, reduce inter- and intra-reader variability, provide more consistent reporting, and can reduce pathologists’ workload significantly. Further improvement to the system will only enhance its performance.

## 5. Conclusions

We have proposed a cell detection and classification system based on a convolutional neural network model for use with the Allred scoring system for breast carcinoma hormone receptor status testing. The system classifies each cell in the ER-stained whole-slide images into negatively, weakly, moderately, and strongly stained cells, before Allred scoring is carried out to recommend hormonal treatment options. To the best of our knowledge, this is the first work of its kind in developing such a system, which would be a valuable tool for histopathologists in improving the reliability of predictive tumor marker reporting as well as reducing manual intervention workload. The cell detection and classification were applied on 40 whole-slide images before Allred scoring was carried out to recommend hormonal treatment options. Experimental result shows very promising observations for both the detection and classification processes, as well as the Allred scoring computation. The automated Allred scoring matches well with pathologists scoring, for both the actual Allred score and hormonal treatment cases. Future work will focus on further improving the accuracy of the system, extending the system on PR and possibly HER2 expression, as well as reducing the computational load in running the system on very large whole-slide breast carcinoma images. The improvement to the accuracy can be either by increasing the training samples, improving the network model, or by proposing some rules for the 0% ER status images as this proved to be the main source of error during performance evaluation.

## Figures and Tables

**Figure 1 diagnostics-12-03093-f001:**
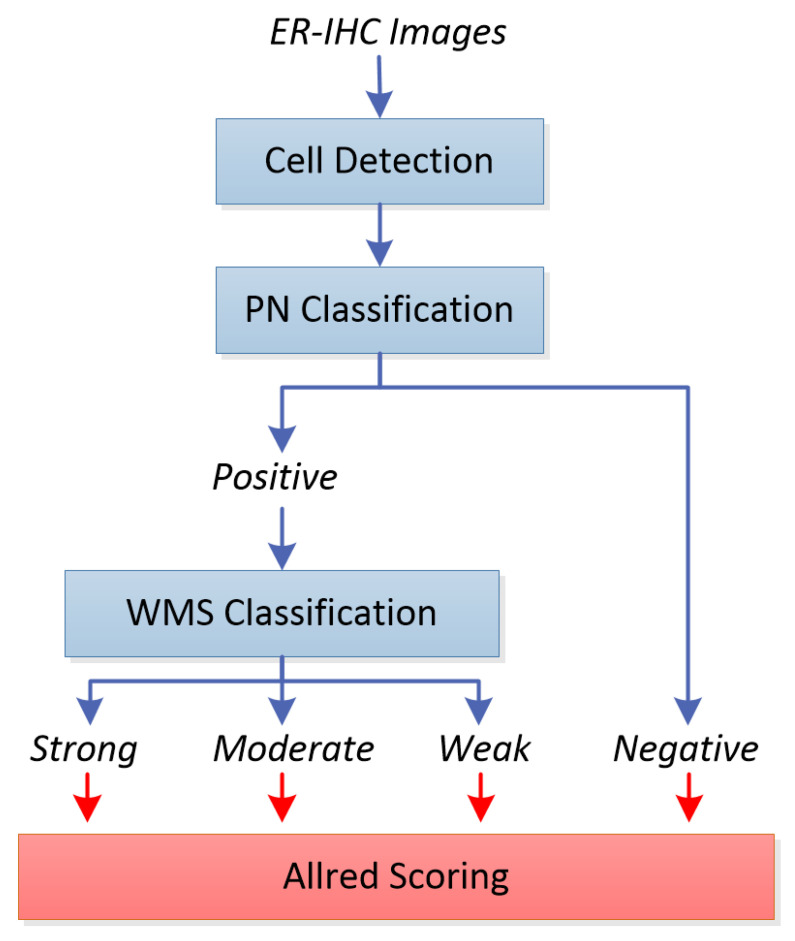
Flowchart for the overview of the system.

**Figure 2 diagnostics-12-03093-f002:**
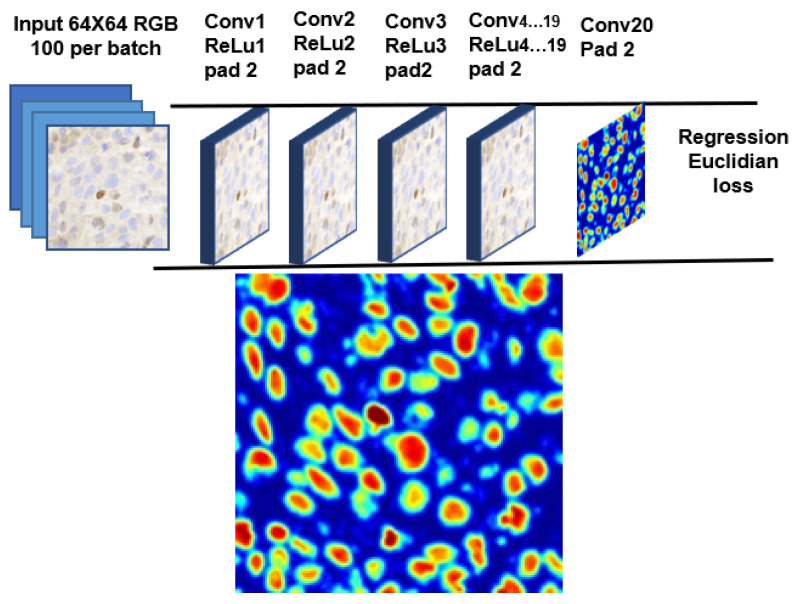
CNN model for cell detection, with regression layer at the end of the network (**top**), and the generated score map at 20× magnification (**bottom**).

**Figure 3 diagnostics-12-03093-f003:**
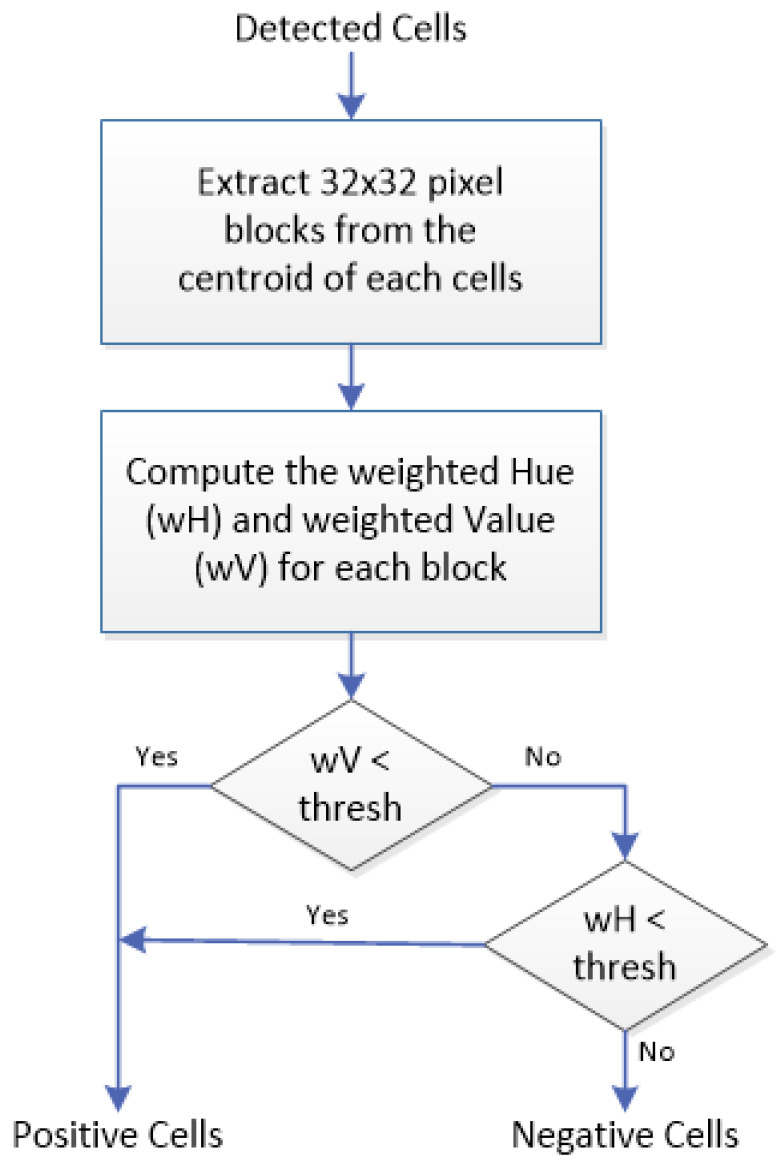
PN classification process.

**Figure 4 diagnostics-12-03093-f004:**
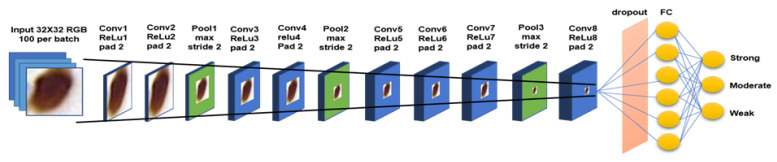
Our proposed 9-layer convolutional layer including the fully connected layer.

**Figure 5 diagnostics-12-03093-f005:**
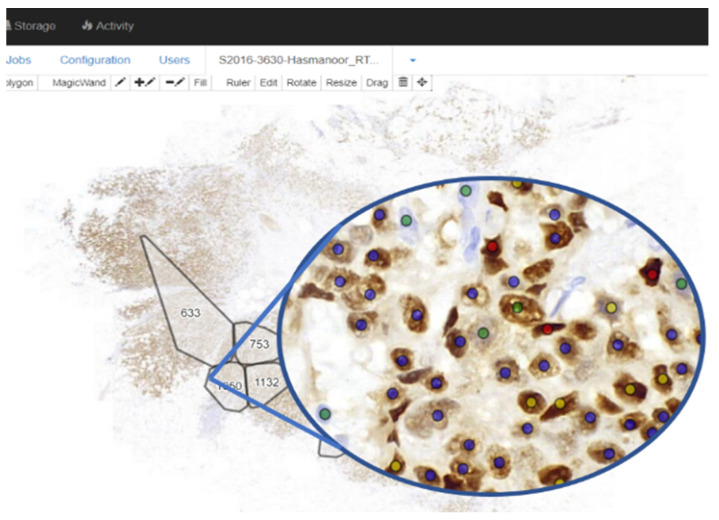
Part of the Cytomine interface, which enabled online annotation on whole-slide images and convenient working on gigabyte data of images. The different color dots show the different classes: green for negative, blue for positive-weak, yellow for positive-moderate and red for positive-strong class nuclei.

**Figure 6 diagnostics-12-03093-f006:**
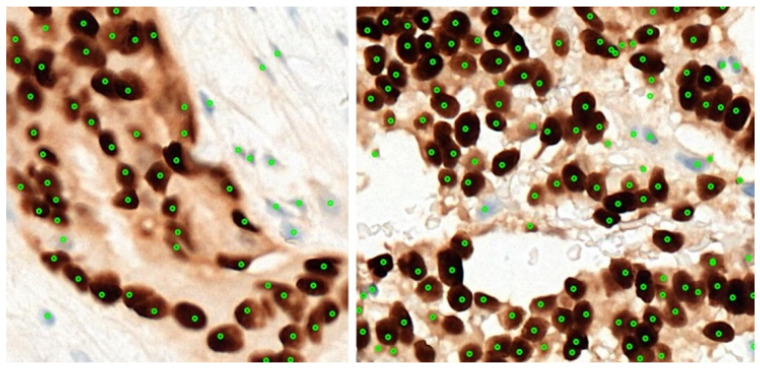
Two of the thirty-seven regions used for the cell detection and classification experiments.

**Figure 7 diagnostics-12-03093-f007:**
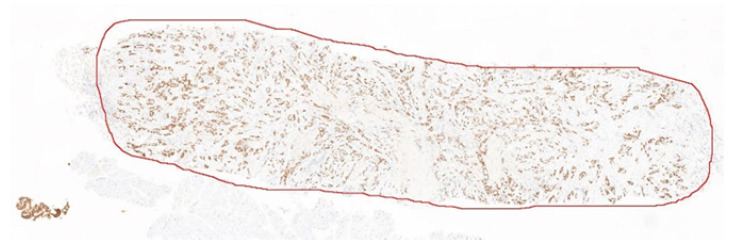
Example of the whole-slide image used in the Allred scoring experiment.

**Figure 8 diagnostics-12-03093-f008:**
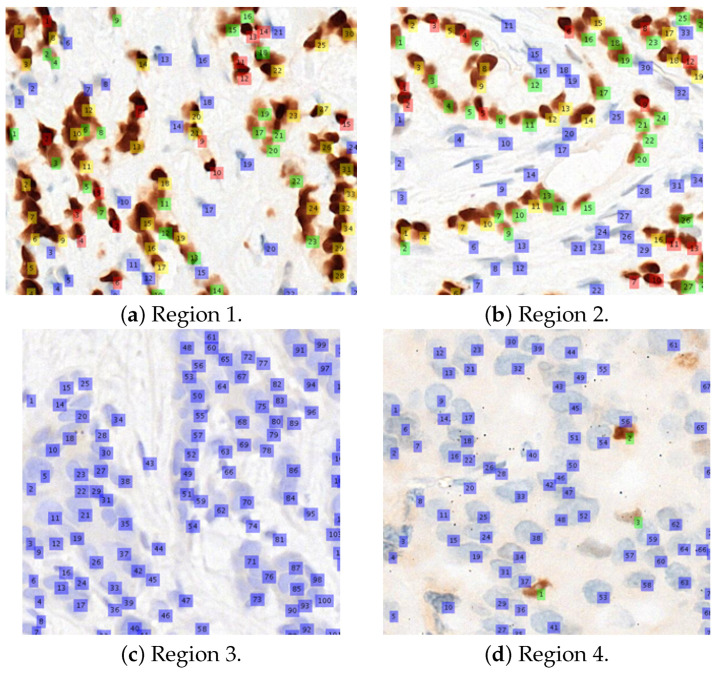
Examples of the classification results for 4 regions. Legends: blue (negative), green (weak), yellow (moderate), and red (strong).

**Figure 9 diagnostics-12-03093-f009:**
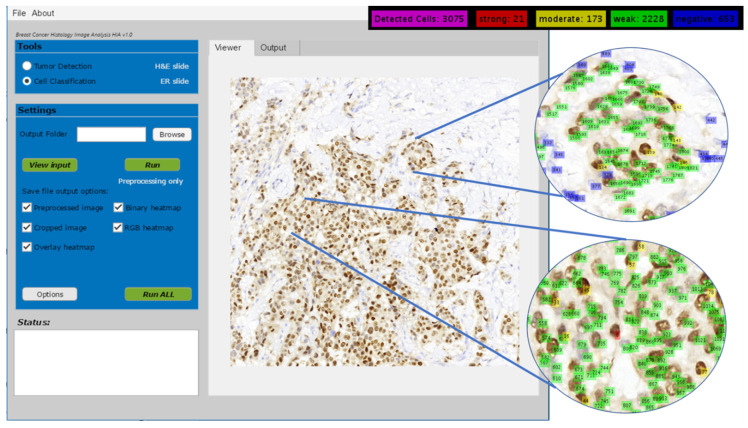
Cell detection and classification in ER-IHC breast carcinoma assessment system.

**Figure 10 diagnostics-12-03093-f010:**
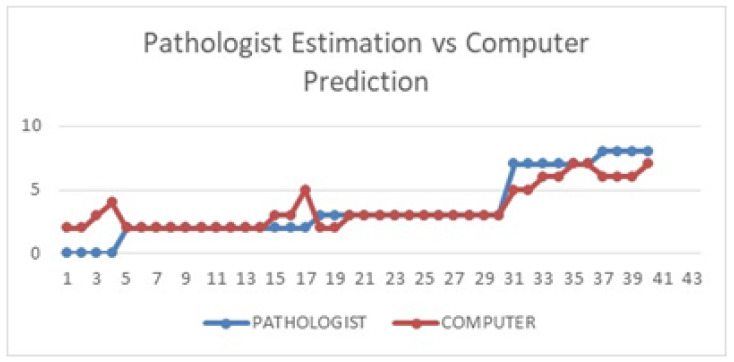
Pathologist’s estimation vs. computer prediction.

**Table 1 diagnostics-12-03093-t001:** Allred score * for estrogen and progestrone receptor evaluation.

ER Status (Positive Cells %)	Proportion Score	Intensity	Intensity Score
0	0	None	0
<1	1	Weak	1
1 to 10	2	Intermediate	2
11 to 33	3	Strong	3
34 to 66	4		
≥67	5		

* Allred score = proportion score + intensity score.

**Table 2 diagnostics-12-03093-t002:** Summary of the images used in the experiments.

Image ID	GT	Exp A	Exp B
01099	Y	Train	Test
05247	Y	Train	Test
05255	Y	Train	Test
05261	Y	Train	Train
05267	Y	Train	Train
05273	Y	Train	Train
05279	Y	Train	Train
05285	Y	Train	Train
05293	N	NA	NA
05299	N	NA	NA
05305	Y	Train	Train
05311	Y	Train	Train
05317	N	NA	NA
05323	Y	Train	Train
05329	Y	Train	Train
05337	Y	Train	Train
05343	Y	Train	Train
05349	Y	Train	Test
05355	Y	Train	Test
05361	Y	Train	Test
05367	Y	Train	Test
05373	Y	Train	Train
05379	Y	Train	Train
05385	Y	Train	Train
05391	Y	Train	Train
05397	Y	Train	Train
05403	Y	Train	Train
05409	Y	Train	Train
05415	Y	Train	Train
05421	Y	Train	Train
05427	Y	Train	Train
05435	Y	Train	Train
05441	Y	Train	Train
05447	Y	Test	Train
05453	Y	Test	Train
05459	Y	Test	Train
05465	Y	Test	Train
60537	Y	Test	Train
55522	Y	Test	Train
78990	Y	Test	Train

**Table 3 diagnostics-12-03093-t003:** Cell detection results.

	Image ID	TP	FP	FN	Recall	Precision
Exp A	05447	107	4	5	0.96	0.96
	05453	100	3	1	0.99	0.97
	05459	43	7	4	0.91	0.86
	05465	73	0	3	0.96	1.00
	60537	112	4	6	0.95	0.97
	55522	103	2	5	0.95	0.98
	78990	99	3	3	0.97	0.97
Exp B	01099	84	13	13	0.87	0.87
	05247	97	0	7	0.93	1.00
	05255	80	15	9	0.90	0.84
	05349	96	6	6	0.94	0.94
	05355	76	0	6	0.93	1.00
	05361	128	0	13	0.91	1.00
	05367	130	5	16	0.89	0.96
	Overall	1328	62	97	0.93	0.96

**Table 4 diagnostics-12-03093-t004:** PN classification results.

	Positive	Negative
Positive	238	12
Negative	1	1077
Non-Tumor	5	57

**Table 5 diagnostics-12-03093-t005:** PN classification results for 14 test images.

	Image ID	All Detected Cells			Only True Tumor Cells		
		Acc.	TPR	TNR	Acc.	TPR	TNR
Exp A	05447	0.96	NA	0.96	1.00	NA	1.00
	05453	0.92	0.00	0.97	0.95	0.00	1.00
	05459	0.86	0.40	0.91	1.00	1.00	1.00
	05465	0.93	0.55	1.00	0.93	0.55	1.00
	60537	0.97	0.00	0.97	1.00	NA	1.00
	55522	0.97	NA	0.97	0.99	NA	0.99
	78990	0.96	0.00	0.97	0.99	0.00	1.00
Exp B	01099	0.87	1.00	0.48	1.00	1.00	1.00
	05247	1.00	1.00	1.00	1.00	1.00	1.00
	05255	0.84	0.99	0.00	1.00	1.00	NA
	05349	0.94	NA	0.94	1.00	NA	1.00
	05355	1.00	1.00	1.00	1.00	1.00	1.00
	05361	0.99	0.93	1.00	0.99	0.93	1.00
	05367	0.96	NA	0.96	1.00	NA	1.00
	Overall	0.95	0.93	0.95	0.99	0.95	1.00

**Table 6 diagnostics-12-03093-t006:** WMS classification results.

	Strong	Moderate	Weak
Strong	48	10	0
Moderate	6	86	7
Weak	0	1	80
Negative	0	0	1
Non Tumor Cells	1	0	4

**Table 7 diagnostics-12-03093-t007:** Accuracy, PPV, and TPR for WMS classification.

	All Detected Positive Cells	Only True Positive Cells
Accuracy	0.88	0.90
PPV-Strong	0.87	0.89
PPV-Moderate	0.89	0.89
PPV-Weak	0.87	0.92
TPR-Strong	0.83	NA
TPR-Moderate	0.87	NA
TPR-Weak	0.99	NA

**Table 8 diagnostics-12-03093-t008:** Allred scoring results.

Image ID	Manual					Automated						
	ER Status	Intensity	P Score	I Score	Allred Score	ER Status	S	M	W	P Score	I Score	Allred Score
05349	0%	NONE	0	0	0	0.83%	0.01	0.02	0.8 *	1	1	2
05367	0%	NONE	0	0	0	0.19%	0.16 *	0.01	0.02	1	3	4
05305	0%	NONE	0	0	0	1.87%	0.02	0.03	1.83 *	2	1	3
55522	0%	NONE	0	0	0	0.8%	0.01	0.03	0.77 *	1	1	2
60537	<1%	WEAK	1	1	2	0.14%	0.02	0.01	0.12 *	1	1	2
05409	<1%	WEAK	1	1	2	0.11%	0.01	0.01	0.09 *	1	1	2
05317	<1%	WEAK	1	1	2	1.29%	0.02	0.04	1.23 *	2	1	3
05343	<1%	WEAK	1	1	2	0.63%	0.02	0.01	0.6 *	1	1	2
78990	<1%	WEAK	1	1	2	0.33%	0.03	0.05	0.25 *	1	1	2
05337	<1%	WEAK	1	1	2	0.80%	0.03	0.04	0.74 *	1	1	2
05403	<1%	WEAK	1	1	2	1.09%	0.03	0.05	1.01 *	2	1	3
05267	<1%	WEAK	1	1	2	0.15%	0.02	0	0.12 *	1	1	2
05385	<1%	WEAK	1	1	2	0.5%	0.07	0.12	0.31 *	1	1	2
05329	<1%	WEAK	1	1	2	0.35%	0.01	0.03	0.32 *	1	1	2
05447	<1%	WEAK	1	1	2	0.56%	0.02	0.01	0.53 *	1	1	2
05459	<1%	WEAK	1	1	2	0.56%	0.03	0.1	2.84 *	1	1	2
05441	<1%	WEAK	1	1	2	47.54%	1.69	2.76	43.09 *	4	1	5
05397	1–10%	WEAK	2	1	3	2.54%	0.02	0.08	2.44 *	2	1	3
05391	1–10%	WEAK	2	1	3	3.55%	0.01	0.06	3.47 *	2	1	3
05355	1–10%	WEAK	2	1	3	1.99%	0.02	0.04	1.93 *	2	1	3
05421	1–10%	WEAK	2	1	3	4.42%	0.05	0.19	4.18 *	2	1	3
05453	1–10%	WEAK	2	1	3	4.61%	0.05	0.07	4.49 *	2	1	3
05279	1–10%	WEAK	2	1	3	3.38%	0.12	0.19	3.07 *	2	1	3
05379	1–10%	WEAK	2	1	3	0.76%	0.02	0.03	0.7 *	1	1	2
05373	1–10%	WEAK	2	1	3	1.31%	0.01	0.02	1.28 *	2	1	3
05361	1–10%	WEAK	2	1	3	2.91%	0.04	0.06	2.81 *	2	1	3
05415	1–10%	WEAK	2	1	3	3.67%	0.03	0.09	3.55 *	2	1	3
05285	1–10%	WEAK	2	1	3	0.22%	0.03	0.05	0.14 *	1	1	2
05427	1–10%	WEAK	2	1	3	7.89%	0.44	0.25	7.2 *	2	1	3
05465	1–10%	WEAK	2	1	3	8.41%	0.01	0.05	8.35 *	2	1	3
05273	>95%	STRONG	5	3	8	46.39%	9.59	19.65 *	17.15	4	2	6
05299	>80%	MODERATE	5	2	7	47.73%	4.97	26.56 *	16.19	4	2	6
05435	>95%	MODERATE	5	2	7	77.44%	12.46	36.19 *	28.8	5	2	7
05247	>90%	MODERATE	5	2	7	49.84%	12.36	17.87 *	19.6	4	1	5
01099	>95%	MODERATE	5	2	7	62.63%	7.55	21.58	33.5 *	4	1	5
05255	>95%	STRONG	5	3	8	73.15%	18.02	40.22 *	14.91	5	2	7
05261	>95%	STRONG	5	3	8	49.47%	14.65	23.91 *	10.91	4	2	6
05311	>95%	MODERATE	5	2	7	73.5%	22.27	40.96 *	10.26	5	2	7
05293	>95%	MODERATE	5	2	7	43.75%	16.66	19.54	7.54	4	2	6
05323	>90%	STRONG	5	3	8	48.39%	11.65	22.11	14.63	4	2	6

* Highest positive class proportion.

## Data Availability

The image data used to support the findings of this study are available from the corresponding author upon request.

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
