# Peer review of "Allred Scoring of ER-IHC Stained Whole-Slide Images for Hormone Receptor Status in Breast Carcinoma"

_diagnostics, 2022, doi:10.3390/diagnostics12123093_

Round 1
Reviewer 1 Report
I reviewed this interesting article which features an innovative automated scoring system to support pathologists' ability to identify breast cancer cases for hormone treatment.
The current results look promising, although some classification issues need to be addressed further.
In my opinion this work is well written and clear in the description of the methodology and in the presentation of the results.
I only suggest further increasing the number of input images in order to strengthen classification accuracy to make this system more acceptable to physicians, who are generally inclined to view automated systems as a threat.
Author Response
Dear Prof,
Please see the attachment.
Thank you.

Reviewer 2 Report
- The authors have yet to discuss much on state-of-the-art on the proposed approach.
- What is the reason for selecting a 32*32 block size for the ROI?
- The authors have used their own dataset. However, to compare it with the existing literature, the authors are suggested to use at least one publicly available and widely used dataset.
- As the data is not larger, why can’t authors try for any machine learning algorithm rather than a deep learning model to reduce the complexity?
- There is no state-of-the-art comparison with the proposed method justify it.
- Add some pros and cons of the proposed approach.
Author Response

(The authors gave the same response as above.)
